# A simplified SARS-CoV-2 detection protocol for research laboratories

**Sean Paz, Christopher Mauer®, Anastasia Ritchie, Janet D. Robishaw, Massimo Caputi®***

Charles E Schmidt College of Medicine, Florida Atlantic University, Boca Raton, Florida, United States of America

* mcaputi@health.fau.edu

## Abstract

Widespread testing is required to limit the current public health crisis caused by the COVID-19 pandemic. Multiple tests protocols have been authorized by the food and drugs administration (FDA) under an emergency use authorization (EUA). The majority of these protocols are based on the gold-standard RT-qPCR test pioneered by the U.S. Centers for Disease Control and Prevention (CDC). However, there is still a widespread lack of testing in the US and many of the clinical diagnostics protocols require extensive human labor and materials that could face supply shortages and present biosafety concerns. Given the need to develop alternative reagents and approaches to provide nucleic-acid testing in the face of heightened demand and potential shortages, we have developed a simplified SARS-CoV-2 testing protocol adapted for its use in research laboratories with minimal molecular biology equipment and expertise. The protocol utilizes TRIzol to purify the viral RNA from different types of clinical specimens, requires minimal BSL-1 precautions and, given its high sensitivity, can be easily adapted to pooling samples strategies.

## Introduction

The first cases of a severe respiratory infection caused by a novel coronavirus were reported in Wuhan, China, in December 2019. As of November 2020, the novel coronavirus SARS-CoV-2 has spread to more than 200 countries with 60 million infected individuals and one and half a million deaths. The US leads the world in both infections and deaths with over 12 million cases and 250,000 deaths [1]. Accurate and rapid diagnostic testing is crucial in reducing community spread and controlling the propagation of the pandemic. Tests should ideally be i) highly sensitive, able to detect mild and asymptomatic infections to facilitate contact tracing, ii) scalable and deployable in millions of units to provide a clear picture of the infection spread and progression in order to facilitate local and national public health policies and iii) highly reliable for monitoring disease progression, remission and aiding in clinical decisions. Estimates of the testing capacity needed to properly monitor and curb the epidemic are in the range of several millions of tests per day in the US alone. Currently in the US, ~1.8 million tests are conducted daily and the number of new positive infections is extremely high (10%) and raising in several states [1].

**Data Availability Statement:** All relevant data are within the manuscript.

**Funding:** This work was supported by the Florida Blue foundation grant "Developing Predictive Algorithms for COVID-19 Infection in FAU Health Care Workers". The funders had no role in study

design, data collection and analysis, decision to publish, or preparation of the manuscript.

**Competing interests:** The authors have declared that no competing interests exist.

Quantitative PCR (qPCR) based molecular tests that detect the presence of the viral nucleic acid offer the most sensitive and reliable method for the detection of SARS-COV-2 in patients' samples. The current gold standard for SARS-CoV-2 detection in the US is the qPCR test developed by the CDC, which utilizes two sets of primers/Taqman probes (2019-nCoV_N1, 2019-nCoV_N2), that detect the sequence coding for the SARS-CoV-2 nucleocapsid (N) gene, and one that detects the cellular gene RNase P (RP) as a control [2]. The products amplified are detected using TaqMan probe fluorescence and a threshold cycle of amplification is set to distinguish positive from negative results. A conclusive positive test result is achieved when both viral targets are amplified, while it is considered negative if none of the viral targets are amplified but the cellular control RNase P is [2].

Testing in the US has been plagued from the beginning by multiple problems. Initially, the development of testing was limited to the CDC and other diagnostic test makers, excluding public health laboratories and academic institutions. Although highly accurate, the CDC approved tests required specialized reagents, equipment, and personnel training. Within a few weeks from the start of the pandemic, US testing centers reported shortages of the reagents utilized in the CDC protocol: from nasopharyngeal swabs to RNA extraction kits to RT-qPCR mix [3, 4]. In the past months, to overcome a health crisis, multiple diagnostic kits have been promptly developed and introduced into the market under a FDA EUA [5]. Unfortunately, since the majority of testing protocols did not undergo the rigorous clinical and scientific evaluation required to receive final FDA approval, their efficacy and sensitivity is, in many cases, less than ideal [6]. At the same time, the test systems currently utilized are not easily scalable to a high-throughput platform to deliver the required millions of tests per day.

Nasopharyngeal (NP) and oropharyngeal (OP) swabs are the most common upper respiratory tract specimen utilized for SARS-CoV-2 diagnostic testing. However, the collection of these specimen types can cause discomfort, bleeding, and requires close contact between healthcare workers and patients, posing the risk of transmission. NP swabs also require the use of specific types of nylon or other synthetic swabs, which are often in short supply, and specialized personnel to safely and properly obtain the sample by inserting the swab through the nasal cavity. The sample needs to be than eluted in viral or universal transport medium (VTM, UTM), stored and transported to a testing facility. Sample collection and storage conditions can dramatically impact the sample quality and subsequent testing steps. The exposure risks to healthcare workers coupled with the invasive nature of these procedures and the global shortages of swabs and personal protective equipment necessitate the development of different diagnostic approaches. Saliva samples have become an increasingly attractive specimen alternative being as- or more sensitive and reliable than NP swabs [7–10]. In addition, saliva samples can be self-obtained and stabilized by the addition of an extraction / preservation reagent as in the protocol we perfected.

In mid-March, after the outbreak had spread to most states, and it became clear that there was a scarcity of testing due to both lack of specialized testing laboratories and supply shortages, our laboratory began optimizing a SARS-CoV-2 testing protocol to overcame the multiple chokeholds in the reagents supply chain. We utilized a simplified TRIzol (guanidinium thiocyanate/phenol-chloroform) RNA extraction method, which is as efficient, or more, than the CDC approved silica-membrane based RNA purification microcolumns in isolating small amounts of viral and cellular RNA from multiple types of samples (nasopharyngeal- oropharyngeal- nasal vestibule- swabs, saliva). We have also shown that samples can be preserved in TRIzol by refrigeration at 4°C for up to a week with little to no loss of viral RNA. The protocol can be easily carried out by any research laboratory equipped with minimal standard equipment, and since saliva can be utilized as a reliable source of virus, samples can be self-obtained by patients and inactivated in TRIzol eliminating the need for medical personnel and higher-

level biosafety protocols and facilities. The protocol we set up is currently utilized in projects monitoring SARS-CoV-2 spread in at risk population categories.

## Results

### A simplified TRIzol protocol for the extraction of viral and cellular RNA

We developed a SARS-CoV-2 testing protocol that utilizes samples from both upper respiratory tract swabs and saliva, which are eluted in TRIzol immediately after collection (Fig 1). The RNA is extracted utilizing a simplified TRIzol protocol for the isolation of minimal amounts of RNA with a recovery rate equal or higher than the commercial sepharose microcolumns. Advantages of this protocol are: i) the use of common chemical reagents that are in abundant supply, ii) the isolation of high quality RNA that can be utilized for multiple assays and RNA sequencing projects, iii) it can be easily adopted by laboratories with minimal equipment and limited molecular biology training and iv) samples in TRIzol can be preserved at 4˚C for over a week with minimal degradation.

### RT-qPCR reliably detects trace amounts of viral RNA

Following its isolation, the viral RNA is reverse transcribed and amplified by qPCR utilizing either a one-step RT-qPCR reaction or separate reverse transcription and qPCR reactions. Commercial master mixes for the RT-qPCR test from ThermoFisher, Quantabio and Promega have been approved by the CDC [2]. However, comparable reagents from a variety of companies including NEB, Applied Biosciences, Roche and Takara have been tested successfully and a growing list of approved alternative commercial reagents can be found at the FDA COVID-19 EUA website [11]. In our assays, we utilized the Agilent Brilliant II RT-qPCR 1-Step Master Mix. The qPCR amplification products are detected by Taqman probes which have a higher specificity and sensitivity when compared to intercalating dyes like SYBR-green. Taqman probes contain a 5′ fluorophore and a 3′ quencher and anneal to sequences within the DNA template generated from the amplification of a target sequence. Taq polymerase degrades the annealed probe and cleaves off the fluorophore, preventing it from being quenched. The fluorescence is proportional to the number of amplified product molecules and can be measured in real-time by a qPCR machine. The CDC protocol utilizes two sets of primers and Taqman probes (nCoV-N1 and nCoV-N2) that specifically anneal the SARS-CoV-2 N gene. A third primer/probe set (RP) targeting the endogenous housekeeping gene RNase P, which is constitutively expressed in the cells present in the donor's sample, is utilized as a positive control of RNA extraction and amplification.

qPCR standard curves were generated for each of the primer-probe sets utilizing 3-fold serial dilutions of purified SARS-CoV-1 genomic RNA (Isolate USA-WA1/2020) and plasmid DNA coding for the RNase P gene (Fig 2). Purified SARS-CoV-1 RNA (Urbani strain) was utilized as a control for the primer's specificity and, as expected, showed no amplification (Fig 5). Both SARS-CoV-2 primer-probe sets were able to amplify as low as 6 copies of viral genomic RNA in less than 34 cycles. This result shows the extreme sensitivity of the CDC primer-probe sets. Following CDC guidelines, samples showing amplification for the nCoV-N1 and nCoV-N2 primer/probe sets with a Ct (amplification cycle where the fluorescence curve exhibits the greatest curvature and exceeds the background fluorescence threshold) of under 35 cycles were considered positive for SARS-COV-2.

### Comparison of three methods for the purification of viral and cellular RNA

The CDC-approved protocols utilize specific RNA extraction kits based on silica-membrane purification microcolumns. TRIzol (Invitrogen) or similar reagents containing guanidinium

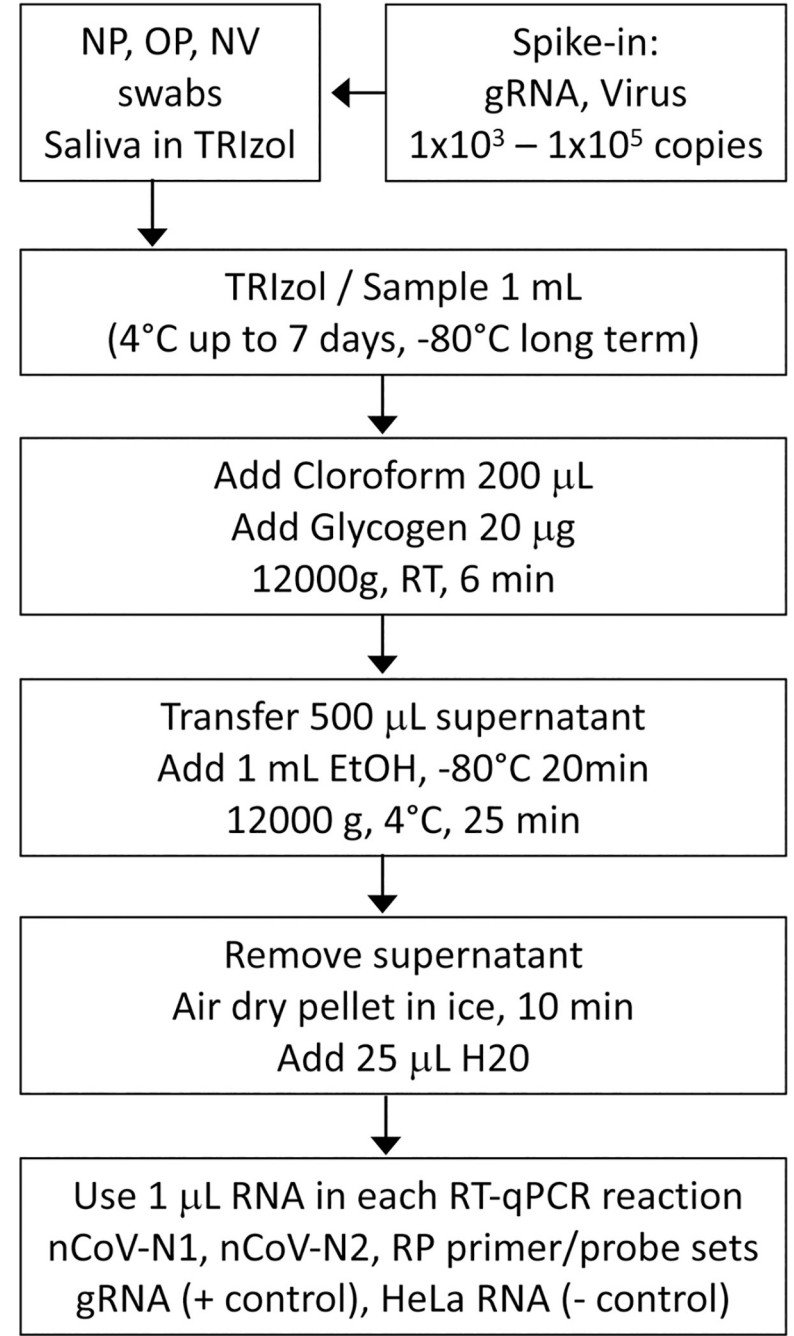

**Fig 1. Overview of the simplified SARS-Cov-2 test protocol.** The RNA is isolated from biological samples (NP, OP, NV or saliva) with a simplified TRIzol RNA isolation protocol. The RT-qPCR reactions utilize the CDC nCoV-N1, nCoV-N2 and RP primer/probe sets. Samples showing amplification for primer/probes nCoV-N1 and nCoV-N2 with a Ct of under 35 cycles are considered positive for SARS-CoV-2. Genomic viral RNA (gRNA) or an inactivated viral preparation can be added to the biological samples from healthy donors to obtain contrived samples.

thiocyanate/phenol-chloroform, have been widely used for the efficient isolation of viral and cellular RNA since 1987 [12]. It is also possible to eliminate RNA isolation altogether by simply lysing the cells via chemical or physical methods. However, RNA isolation improves detection

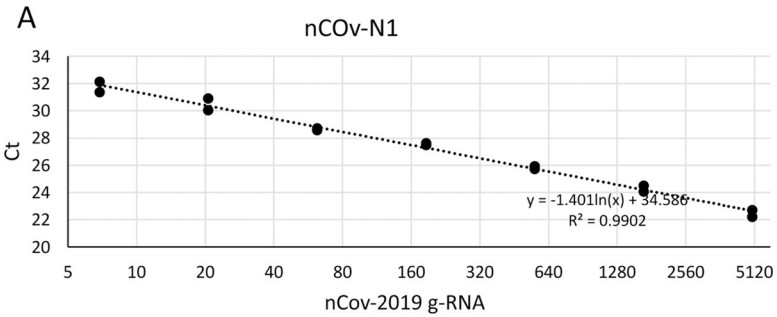

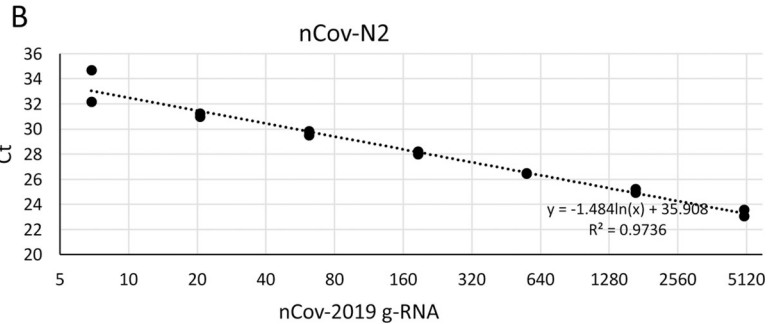

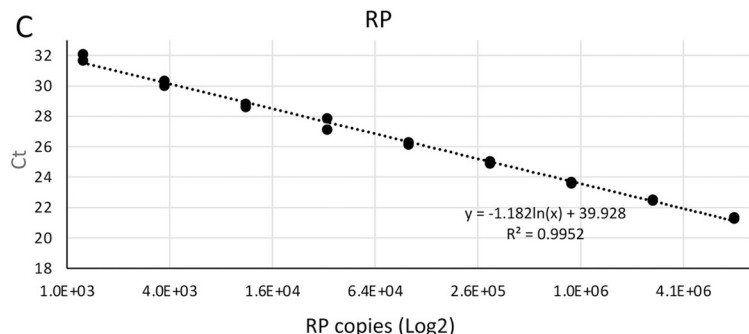

**Fig 2. SARS-CoV-2 and RNAse P primer/probes RT-qPCR standard curves.** Primers-probes sets A) nCoV-N1 and B) nCoV-N2 standard curves were generated by amplifying 3-fold serial dilutions of purified viral genomic RNA (BEI resources, NR52285). 6 copies of viral genomic RNA were detected with a Ct lower than 34 cycles with both primer/probe sets. C) A titration curve for the endogenous housekeeping gene RNase P probe set (RP) was obtained utilizing plasmid DNA coding for the RNase P gene.

sensitivity and might be required to remove compounds that may inhibit the reverse transcription or amplification steps.

We compared the simplified TRIzol extraction protocol with the RNAqueous Total RNA Isolation micro-Kit, which utilizes glass fiber filter microcolumns for the isolation of RNA from small amounts of sample, and a rapid hypotonic buffer freeze protocol for rapid RT-qPCR assays without RNA isolation [13]. We utilized varying amounts of Hela cells spiked-in with a constant amount of viral genomic RNA (10,000 copies) to compare the efficiency of RNA recovery from limited amounts of sample in the three protocols (Fig 3). We observed a higher viral RNA recovery efficiency utilizing the TRIzol method. This was more evident when limiting amounts of cells were present in the sample.

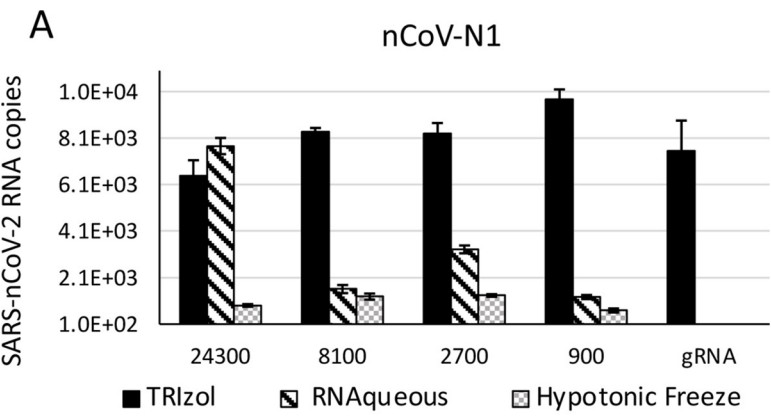

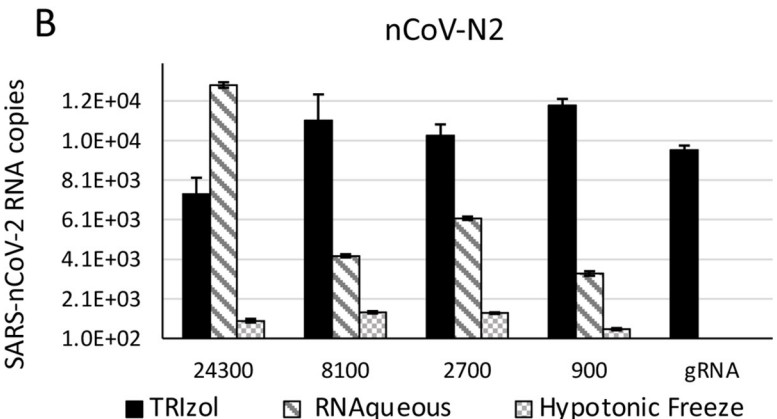

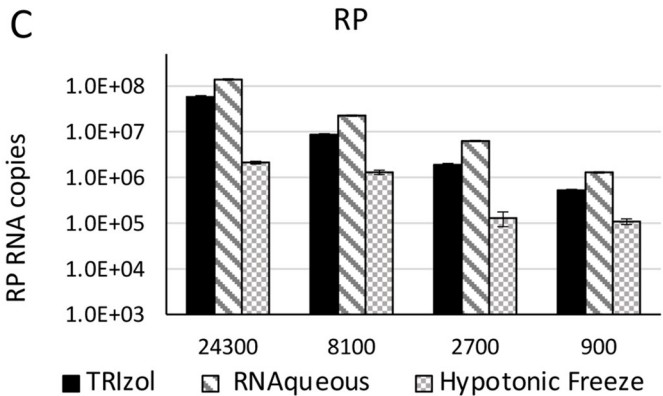

**Fig 3. SARS-CoV2 different RNA isolation methods.** 10,000 copies of viral genomic RNA (gRNA) were added to a serial dilution of HeLa cells (from 24,300 to 900). Total RNA was isolated with TRIzol, the RNAqueous kit and the hypotonic freeze method. 1 μL of each RNA preparation was amplified utilizing the A) nCoV-N1, B) nCoV-N2 and C) RP primer sets. Following isolation, the RNA was resuspended in 25 μL of ddH$_2$O. Isolation of the RNA with 100% efficiency will yield 400 copies / μL of gRNA. 400 copies of purified gRNA were directly amplified as a control for RNA isolation efficiency.

## Detection of SARS-CoV-2 in different types of clinical specimens

Saliva specimens can be provided easily by the patients and have been shown to have a concordance rate of greater than 95% with NP specimens in the detection of respiratory viruses, including coronaviruses [7, 14, 15]. Swabs of the nasal vestibule (NV) can also be utilized as a sample source although their concordance rate with NP samples is only 80% [16]. To evaluate the simplified TRIzol method in isolating viral RNA from NP, OP, NV and saliva samples, we utilized contrived samples from 7 healthy donors spiked-in with a preparation of inactivated SARS-CoV-2 virus (isolate USA-WA1/2020) containing 10,000 viral genome copies (Fig 4). This viral load is at the lower end of what is normally observed in patients [17–19]. The amount of viral RNA recovered was calculated with a liner interpolation based on the standard curves shown in Fig 1. Overall, the recovery efficiency of viral RNA from saliva was comparable or better than the one from the NP and NV samples, but it was lower in OP samples. Notably, the amount of RP control RNA recovered from the OP samples was roughly 3 orders of magnitude lower than that recovered from the saliva, NP and NV samples. This suggests the presence of less epithelial cells in the OP samples and possibly the presence of enzymes or impurities that might interfere with the stability of both viral and cellular RNA or with some of the steps in the RNA purification / RT-qPCR protocol.

## Contrived saliva samples require the addition of the inactivated virus

Contrived samples are commonly utilized by testing manufacturers to evaluate the performance of new diagnostics [11] and have the advantage of containing known amounts of viral

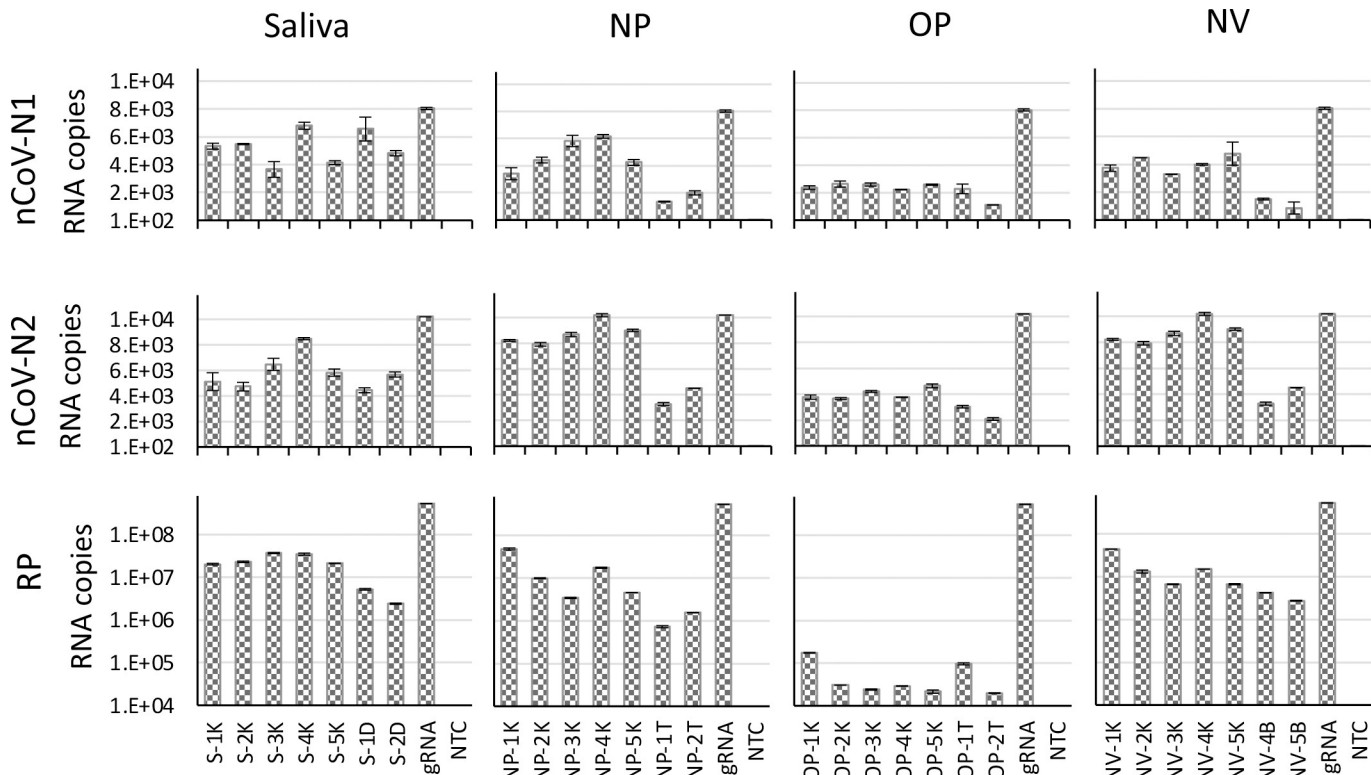

**Fig 4. Detection of SARS-CoV-2 in contrived samples from the upper respiratory tract.** NP, OP, NV and saliva samples from 7 healthy donors were spiked-in with a preparation of inactivated SARS-CoV-2 virus (isolate USA-WA1/2020) containing 10,000 viral genome copies. RNA extracted with the simplified TRIzol method was amplified with the nCoV-N1, nCoV-N2 and RP primer/probe sets. 400 copies of purified gRNA were directly amplified as a control for RNA isolation efficiency.

RNA, thus allowing for a precise evaluation of the test sensitivity. Nevertheless, if purified viral RNA is directly added to the sample, the RNases present in the biological specimen can quickly degrade the unprotected RNA. This is not true for the RNA present within the virion. In Fig 5, we show that the addition of purified SARS-CoV-2 genomic RNA to a saliva sample before addition of TRIzol greatly affects RNA recovery, while the viral RNA is efficiently recovered when it is added to the sample after the TRIzol. Thus, viral preparations containing the

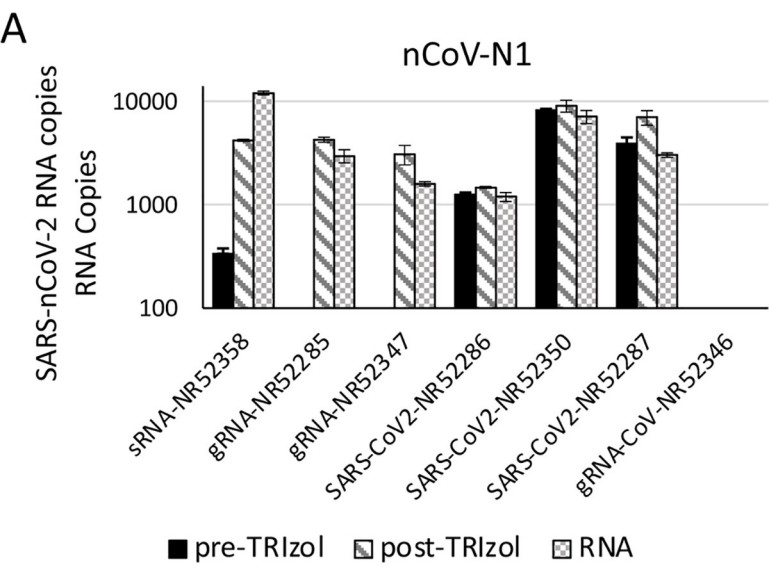

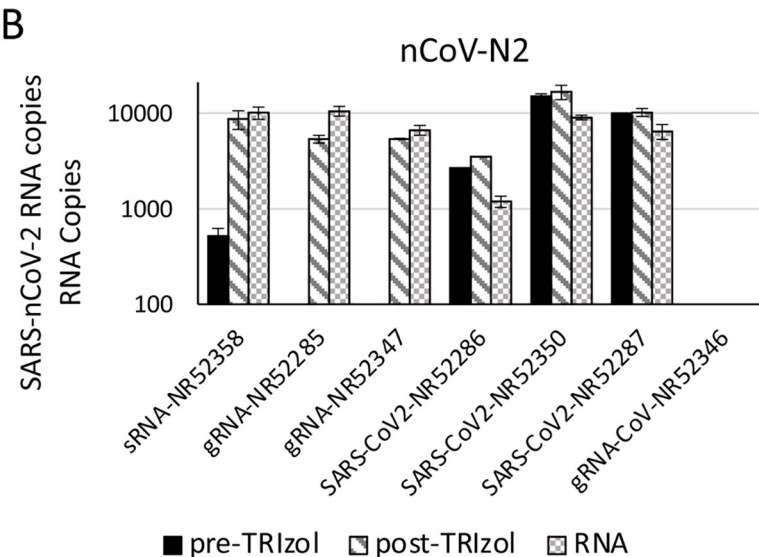

**Fig 5. SARS-CoV2 gRNA degradation in saliva samples.** 10,000 copies of synthetic SARS-CoV-2 RNA (NR52286), purified SARS-CoV-2 RNA (NR52358, (NR52285, NR52347), inactivated viral preparations (NR52350, NR52287), and a control SARS-CoV-1 genomic RNA (NR52346) were added before (pre-) and after (post-) addition of TRIzol to a saliva sample or directly to the TRIzol sample without saliva (RNA). RNA was isolated with the simplified TRIzol protocol and amplified with the A) nCoV-N1 and B) nCoV-N2 primer sets.

inactivated virus can be safely utilized to obtain contrived samples that undergo minimal RNA degradation before the addition of TRIzol. A preparation of heat-inactivated or gamma-irradiated SARS-CoV-2 is preferable to purified genomic RNA in obtaining contrived samples especially when biological specimens containing considerable quantities of RNases, such as saliva, are utilized.

### Samples preserved in TRIzol are stable at 4˚C

In the most commonly used COVID-19 testing protocols, a health care provider collects a NP or OP swab and transfers it to a vial containing a few milliliters of VTM. The sample is then transported to a laboratory for testing. The transport and storage can take from a few hours to a few days depending on the distance and processing times of the nearest clinical laboratory. The CDC recommends that specimens are stored at 2–8˚C for up to 72 hours after collection and at -70˚C or lower for longer periods of time [2]. However, the logistics of having multiple sample collection points, chokeholds in the reagents supply chain, and abrupt increases in the demand for testing due to local outbreaks might generate unexpected delays in processing the samples. TRIzol offers a distinct advantage over the standard VTM. In Fig 6, we show that samples spiked-in with an inactivated SARS-CoV-2 preparation are stable when preserved at 4˚C for a period of 7 days, while only 10% of the genomic RNA can be recovered after being preserved in VTM. Surprisingly, the cellular control RNase P RNA appears to be degraded more efficiently than the viral RNA while in TRIzol, suggesting an inherent higher stability of the viral RNA.

## Discussion

In response to the global COVID-19 emergency and to address the need for increased testing over the past few months, researchers across disciplines have quickly compared widely available commercial products and testing protocols, and repurposed existing reagents and infrastructure creating novel solutions to optimize the COVID-19 testing pipeline. We give a detailed description of a testing protocol for detection of minimal quantities of SARS-CoV-2 that can be utilized by most laboratories equipped with standard molecular biology equipment without the need of higher biosafety facilities. The use of saliva as a sample source and TRIzol as a transport/lysis buffer allows for self-collection, minimizes the chance of exposing healthcare workers and allows the preservation of the sample in standard refrigeration conditions for up to a week with no loss of viral RNA integrity.

The virus has been shown to be present at high titer in saliva [7, 8, 10, 20]. In addition, we have observed a recovery of viral RNA between 50 and 90% with the TRIzol method and the RT-qPCR assay we utilize can amplify as little as 6 copies of viral RNA. The high sensitivity of this protocol might be useful in testing patients with low viral titers such as asymptomatic patients [19] or prior to quarantine release. Furthermore, several patient samples can be pooled to decrease the number of tests required for larger populations [21, 22]. In this approach the samples are first pooled and tested, positive pools are than retested individually. This is a relatively simple solution, which decreases the testing resources used but results in a loss in sensitivity from diluting positive patient samples with negative ones, hence the need of highly sensitive tests that utilize biological materials, like saliva, which can be obtained in larger amounts and can be easily preserved for re-testing.

A high demand for testing can be expected in the foreseeable future as testing of the general population and asymptomatic individuals becomes more widespread. The lack of control of the pandemic in many underdeveloped countries and the second wave of viral outbreak in the fall 2020 are also compelling reasons to increase the testing efforts. We are hopeful that a

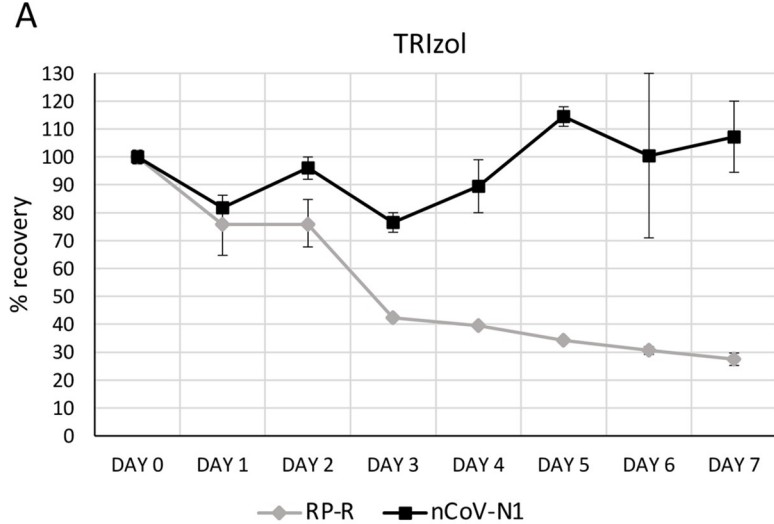

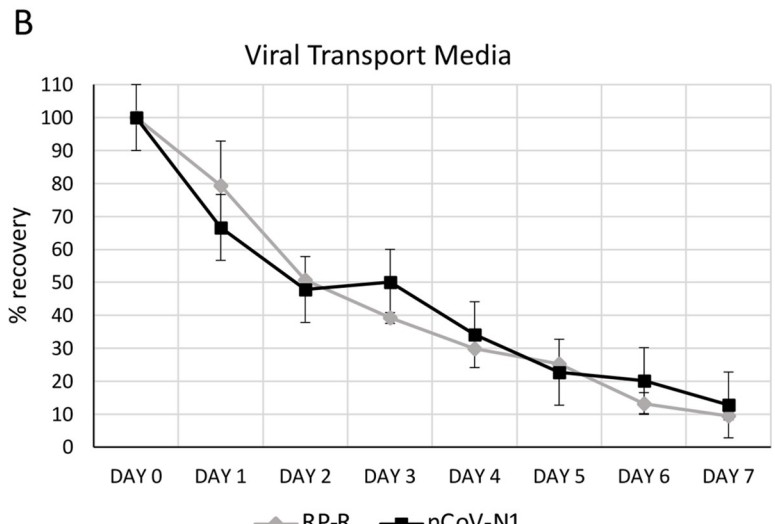

**Fig 6. Stability of SARS-CoV2 and cellular RNA in transport media and TRIzol.** A time course was set up with samples containing 24,000 HeLa cells and a preparation of inactivated SARS-CoV-2 virus (isolate USA-WA1/2020) containing 10,000 viral genome copies. Samples were preserved in either A) TRIzol or B) viral transport media (VTM) at 4˚C for up to 7 days. RNA was isolated with the simplified TRIzol protocol and amplified with the nCoV-N1 and RP primer/probe sets.

combination of testing approaches, including protocols like ours, may be the most efficient way to fill the current and future gaps in testing.

## Materials and methods

### Sample collection and storage

NP, OP and NV specimen were obtained utilizing nylon swabs and by placing the swabs immediately into sterile tubes containing either 2 mL of viral transport media (VTM) or 2 mL of TRIzol (Invitrogen). Samples are vortexed to elute the tissue and viral particles. Samples in VTM can be stored at 2–8˚C for up to 72 hours after collection. If a delay in extraction is expected, specimens should be stored at -70˚C. Samples in TRIzol are stable at 4˚C for at least

a week, and longer storage can be carried out at -20˚C. Saliva samples should be produced early in the morning from the posterior oropharynx (ie, coughed up by clearing the throat) before toothbrushing and breakfast. Nasopharyngeal secretions move posteriorly, and bronchopulmonary secretions move by ciliary activity to the posterior oropharyngeal area during sleep. 1 mL of saliva should be placed in a tube containing 2 mL of TRIzol.

## RNA isolation

TRIzol RNA extraction was carried out from a biological sample/TRIzol mixture as follows: transfer 1 mL of the TRIzol/sample mixture to a 1.5 mL microfuge tube adding 200 μL chloroform and 1 μL glycogen (RNA grade, 20 μg/μL). The sample is then vortexed for 5 seconds and centrifuged at 12,000g at RT for 6 minutes. Transfer 500 μL of supernatant with a p200 pipette to a new tube being careful not to disturb the interface. Add 1 mL of ethanol and place at -80˚C for 20 minutes. Centrifuge at 12000g at 4˚C for 25 minutes. Remove the supernatant carefully without dislodging the pellet (often not visible) and air dry while on ice for 10 min. Resuspend in 25 μL of RNase free ddH2O. RNA extraction was carried out utilizing the hypotonic buffer and freeze protocol as previously described [13]. Briefly, cells are pelleted and resuspended in 50 μL of hypotonic freezing buffer (75 mM NaCl, 10 mM Tris, pH 8.0, 2.5mM DTT). Freeze immediately in a box containing 95% ethanol chilled to -80˚C for 3 minutes. Thaw and repeat. Vortex the sample briefly and centrifuge at 12000g at 4˚C for 3 minutes. Transfer the supernatant to a new tube. RNA extraction was carried out utilizing the RNAqueous-Micro Kit (Ambion-Invitrogen) following the manufacturer instructions. RNA samples were stored at -80˚C.

## RT-qPCR assay

RT-qPCR assays were carried out utilizing the Agilent qRT-PCR Brilliant II Probe Master Mix (Cat #600809) and an AriaMx Real-time PCR System. 1 μL of the RNA preparation was loaded in each 25 μL RT-qPCR reaction. The RT-qPCR was performed as follows: 50˚C 30 min, 95˚C 10 minutes (RT step, 1 cycle) followed by 95˚C 15 seconds, 60˚C 1 minute (40 cycles, amplification). Reactions were set up with the following primer sets and concentrations following CDC guidelines [2]: 2019-nCoV_N1-F 5'-GACCCCAAAATCAGCGAAAT-3' (400 nM final), 2019-nCoV_N1-R 5'-TCTGGTTACTGCCAGTTGAATCTG-3' (400 nM final), 2019-nCoV_N1-P 5'-FAM-ACCCCGCATTACGTTTGGTGG ACC-BHQ1-3' (100 nM final); 2019-nCoV_N2-F 5'-TTACAAACATTGGCCGCAAA-3' (400 nM final), 2019-nCoV_N2-R 5'-GCGCGACATTCCGAAGAA-3' (400 nM final), 2019-nCoV_N2-P 5'-FAM-ACAATTTGC CCCCAGCGCTTCAG-BHQ1-3' (100 nM final); RP-F 5'-AGATTTGGACCTGCGAGCG-3' (400 nM final), RP-R 5-GAGCGGCTGTCTCCACAAGT-3'(400 nM final), RP-P 5'-FAM–TTCTGACCTGAAGGCTCTGCGCG–BHQ1-3' (100 nM final). All the primers can be obtained premixed in the proper proportions in the Integrated DNA Technologies (IDT) RUO Kit (Cat#10006713). Samples showing amplification for primer/probes nCoV-N1 and nCoV-N2 with a Ct of under 35 cycles are considered positive for SARS-COV-2. Samples with no Ct for the nCoV-N1, nCoV-N2 and no Ct for the RP cellular RNA control are considered not conclusive and should be re-processed. All genomic RNAs and viral preparations were obtained from BEI resources (https://www.beiresources.org/). The following reagents can be utilized as positive controls for SARS-CoV-2: NR-52358 (synthetic RNA for N, E genes), NR-52285 (purified genomic RNA), NR-52347 (purified genomic RNA), NR-52350 (purified genomic RNA and cellular RNA), NR-52286 (heat inactivated SARS-CoV-2), NR-52287 (gamma irradiated SARS-CoV-2). As negative control we utilized: NR-52349 (purified genomic RNA, SARS-CoV-1, Urbani strain), NR-52346 (SARS-CoV-1, Urbani strain). As a positive

control for the cellular control RNase P and as an added negative control for the SARS-CoV-2 primer/probe sets, we utilized RNA extracted from HeLa cells (ATCC). All thermal cycling data are representative of two independent experimental replica and two technical replicates.

The samples utilized have been de identified and classified as non human research by the Florida Atlantic University Health Sciences IRB. The Florida Atlantic University Health Sciences IRB has published the following Board Document on IRBNet: Project Title: [1584869–1] COVID-19 Testing in relation to this project.

## Author Contributions

**Conceptualization:** Janet D. Robishaw, Massimo Caputi.

**Data curation:** Massimo Caputi.

**Formal analysis:** Massimo Caputi.

**Funding acquisition:** Janet D. Robishaw, Massimo Caputi.

**Investigation:** Sean Paz, Christopher Mauer, Anastasia Ritchie.

**Methodology:** Sean Paz, Massimo Caputi.

**Project administration:** Massimo Caputi.

**Supervision:** Massimo Caputi.

**Validation:** Christopher Mauer.

**Writing – original draft:** Massimo Caputi.

**Writing – review & editing:** Sean Paz.

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
