## [Decision Letter · Decision Letter 0]

20 Nov 2020

PONE-D-20-26723

A simplified SARS-CoV-2 detection protocol for research laboratories

PLOS ONE

Dear Dr. Caputi,

Thank you for submitting your manuscript to PLOS ONE. After careful consideration, we feel that it has merit but does not fully meet PLOS ONE’s publication criteria as it currently stands. Therefore, we invite you to submit a revised version of the manuscript that addresses the points raised during the review process.

The comments raised from the three independent reviewers have diverged. Please consider their comments very carefully.

We look forward to receiving your revised manuscript.

Kind regards,

Etsuro Ito

Academic Editor

PLOS ONE

Journal Requirements:

Reviewers' comments:

Reviewer's Responses to Questions

**Comments to the Author**

1. Is the manuscript technically sound, and do the data support the conclusions?

Reviewer #1: Yes

Reviewer #2: No

Reviewer #3: Yes

2. Has the statistical analysis been performed appropriately and rigorously? 

Reviewer #1: N/A

Reviewer #2: No

Reviewer #3: Yes

3. Have the authors made all data underlying the findings in their manuscript fully available?

Reviewer #1: Yes

Reviewer #2: Yes

Reviewer #3: Yes

4. Is the manuscript presented in an intelligible fashion and written in standard English?

Reviewer #1: Yes

Reviewer #2: Yes

Reviewer #3: Yes

5. Review Comments to the Author

Reviewer #1: The manuscript entitled, "A simplified SARS-CoV-2 detection protocol for research laboratories" provide the detailed procedure of extraction of SARS-CoV2 RNA from different sample types. It is a carefully designed study and well written except for a few typos. After revising the minor English typos, should be accepted.

Reviewer #2: The central theme of the paper is reporting a simplified test requiring less biosafety precautions for diagnosis of SARS-COV-2 virus. There is nothing new in this paper because RNA extraction using the Trizol is known long before. The novelty in the paper is not impressive and it does not reveal high scientific value. The paper is not recommended for publication.

Reviewer #3: Report on the manuscript # D-20-26723 by Paz et al. entitled “A simplified SARS-CoV-2 detection protocol for research laboratories”

COMMENTS TO AUTHORS:

The manuscript examined a TRIzol-based protocol for detection SARS-CoV-2 using NP, OP, NV or saliva as a starting material. The authors spiked the samples with viral RNA and treated them with TRIzol for detection of SARS-CoV-2 using qPCR. The method needs a BSL-1 molecular facility and compares three different viral RNA purification procedures with higher yields and more stability with the TRIzol method. The manuscript is well written and the text flows well with minor editorial and style issues. Here are more specific comments that need to be addressed.

Specific comments:

1. The abstract has no mention of the method or protocol of the study.

2. If possible, the first paragraph of the introduction needs to be updated with more recent epidemiologic data regarding the prevalence and mortality rates.

3. The Results section contains a lot of background and discussion information. It is preferable to either join the Results and Discussion sections or move the non-results text either to the introduction or the Discussion section.

Minor issues:

1. Acronyms e.g. CDC (P1 L39) should be spelled out and abbreviated at their first appearance and then used as abbreviated later in the manuscript.

2. P5 L86, “Figure 1 and 4” should be “Figures 1 and 4” and I suggest numbering the figures in the sequence they appear in the manuscript text. The authors refer to figures 1, 4, 3 and 5 in this order in the text. It should be 1, 2, 3, 4…etc.

3. L155 states the standard curve in Figure 1 while the standards curves are shown in figure 2.

4. The centrifugation process is sometimes stated in “g” and sometimes in “rmp” e.g. P12 L228 and 236. It should be consistent.

5. It is preferrable to number and refer to the different panels in the figures with multiples panels (Figures 2-6) in the figure legends. Also, what does the error bars represent in the figures particularly Figure 4 with attention to what the Y axis label on that figure. In addition, Figure 6 legend shows VTM while it is UTM on the lower panel!!!

6. PLOS authors have the option to publish the peer review history of their article (what does this mean?). If published, this will include your full peer review and any attached files.

Reviewer #1: No

Reviewer #2: No

Reviewer #3: **Yes: **Sayed F Abdelwahab

---

## [Author Response · Author response to Decision Letter 0]

24 Nov 2020

Response to Reviewers:

In the marked revised version of the manuscript the modifications we highlighted in yellow the text modified in response to the reviewer’s comments. 

Reviewer #1

As suggested by reviewer #1 we revised minor typos throughout the manuscript

Reviewer #2

There were no specific critiques to address

To the question: Is the manuscript technically sound, and do the data support the conclusions?

The reviewer answered NO but no specific reasons were given. No specific reasons were given on the questions Has the statistical analysis been performed appropriately and rigorously? Answer No.

As for the comment “There is nothing new in this paper because RNA extraction using the Trizol is known long before. The novelty in the paper is not impressive and it does not reveal high scientific value.” We appreciate the comment of the reviewer although PLOSone evaluate submitted manuscripts on the basis of methodological rigor and high ethical standards, regardless of perceived novelty. Nevertheless the application of a shortened TRIzol RNA extraction protocol to nCoV2 samples from multiple clinical sources had not been described in details at the time of our submission, nor it was described the effect of TRIzol in preserving specimen at 4C or the fact that viral preparation and not viral RNA should be used for contrived samples in this type of studies. The fact that we have utilized the protocol here described to test successfully test over 1000 primary care workers (100% sensitivity and specificity to date) and I have received requests for this detailed protocol by a dozen other labs is confirming the overall quality of the work we carried out in setting up a useful nCoV2 testing protocol. 

Reviewer #3

As suggested by the reviewer we updated the epidemiological data in the introduction. 

We added the method utilized in the abstract. 

We moved two descriptive paragraphs from the results to the introduction (from sub-heading: A simplified TRIzol protocol for the extraction of viral and cellular RNA and from sub-heading: Detection of SARS-CoV-2 in different types of clinical specimens). 

We added a subheading to better organize the results (Contrieved saliva samples require the addition of the inactivated virus.). 

We corrected the text and the figures as suggested by the reviewer.

---

## [Decision Letter · Decision Letter 1]

8 Dec 2020

A simplified SARS-CoV-2 detection protocol for research laboratories

PONE-D-20-26723R1

Dear Dr. Caputi,

We’re pleased to inform you that your manuscript has been judged scientifically suitable for publication and will be formally accepted for publication once it meets all outstanding technical requirements.

Kind regards,

Etsuro Ito

Academic Editor

PLOS ONE

Reviewers' comments:

Reviewer's Responses to Questions

**Comments to the Author**

1. If the authors have adequately addressed your comments raised in a previous round of review and you feel that this manuscript is now acceptable for publication, you may indicate that here to bypass the “Comments to the Author” section, enter your conflict of interest statement in the “Confidential to Editor” section, and submit your "Accept" recommendation.

Reviewer #3: (No Response)

2. Is the manuscript technically sound, and do the data support the conclusions?

Reviewer #3: Yes

3. Has the statistical analysis been performed appropriately and rigorously? 

Reviewer #3: No

4. Have the authors made all data underlying the findings in their manuscript fully available?

Reviewer #3: Yes

5. Is the manuscript presented in an intelligible fashion and written in standard English?

Reviewer #3: Yes

6. Review Comments to the Author

Reviewer #3: (No Response)

7. PLOS authors have the option to publish the peer review history of their article (what does this mean?). If published, this will include your full peer review and any attached files.

Reviewer #3: No

---

## [Editor Report · Acceptance letter]

11 Dec 2020

PONE-D-20-26723R1 

A simplified SARS-CoV-2 detection protocol for research laboratories 

Dear Dr. Caputi:

I'm pleased to inform you that your manuscript has been deemed suitable for publication in PLOS ONE. Congratulations! Your manuscript is now with our production department. 

Kind regards, 

on behalf of

Prof. Etsuro Ito 

Academic Editor

PLOS ONE